# Expression of EMP 1, 2, and 3 in Adrenal Cortical Neoplasm and Pheochromocytoma

**DOI:** 10.3390/ijms241613016

**Published:** 2023-08-21

**Authors:** Yoon Jin Cha, Ja Seung Koo

**Affiliations:** Department of Pathology, Yonsei University College of Medicine, Seoul 120-752, Republic of Korea; yooncha@yuhs.ac

**Keywords:** adrenal gland tumor, adrenal cortical neoplasm, epithelial membrane proteins, pheochromocytoma

## Abstract

The purpose of this study is to investigate the expression of the epithelial membrane proteins (EMP) 1, 2, and 3 in adrenal gland neoplasm and to explore the broader implications of this. Tissue microarrays were constructed for 132 cases of adrenal cortical neoplasms (ACN) (adrenal cortical adenoma (115 cases), and carcinoma (17 cases)) and 189 cases of pheochromocytoma. Immunohistochemical staining was performed to identify EMP 1, 2, and 3, and was compared with clinicopathological parameters. The H-score of EMP 3 (*p* < 0.001) was higher in pheochromocytoma when compared to that of ACN, and the H-score of EMP 1 (*p* < 0.001) and EMP 3 (*p* < 0.001) was higher in adrenal cortical carcinomas when compared to that of adrenal cortical adenomas. A higher EMP 1 H-score was observed in pheochromocytomas with a GAPP score ≥3 (*p* = 0.018). In univariate analysis, high levels of EMP 1 and EMP 3 expression in ACN were associated with shorter overall survival (*p* = 0.001). Differences were observed in the expression of EMPs between ACN and pheochromocytoma. EMPs are associated with malignant tumor biology in adrenal cortical neoplasm and pheochromocytoma, suggesting the role of a prognostic and/or predictive factor for EMPs in adrenal tumor.

## 1. Introduction

Adrenal gland neoplasms primarily encompass adrenal cortical neoplasms (ACNs) and pheochromocytomas (PCCs), localized in the adrenal cortex and adrenal medulla, respectively. ACNs, comprising adrenal cortical adenomas (ACAs) and adrenal cortical carcinomas (ACCs), are relatively uncommon tumors, and distinguishing between them histologically presents challenges [1]. ACCs, characterized by their rarity and high malignancy, still possess significant gaps in our understanding of their tumor biology, with a lack of effective targeted therapies available [2]. ACC patients are known to have a recurrence rate of over 50% within five years after surgical resection [3], and a significant number of patients are already diagnosed with metastatic disease at the time of diagnosis [4,5]. Therefore, to date, complete surgical excision remains the most important treatment modality. However, additional medical treatment can play a crucial role in ACC management. In addition to traditional chemotherapy agents, the only ACC medication approved by the US FDA is mitotane, which inhibits steroid synthesis [6]. Furthermore, clinical trials are underway for immunotherapy agents [7,8,9,10], tyrosine kinase inhibitors [11,12], and monoclonal antibody drugs [13,14]. However, their effectiveness is still limited.

PCC is commonly discovered incidentally through health screenings, accounting for approximately 20–60% of cases, and it represents about 4–8% of all adrenal incidentalomas [15,16]. Previously, about 10% of PCCs were considered malignant [17]. However, currently, all PCCs are believed to have metastatic potential [18]. Most PCCs can be treated surgically [19]. For metastatic PCC, traditional chemotherapy agents have been used in the past [20,21]. However, more recent attempts involve the use of tyrosine kinase inhibitors [22], peptide receptor radionuclide [23], histone deacetylase inhibitors [24], and immune checkpoint inhibitors [25]. Likewise, it is challenging to differentiate between benign and malignant PCC through histological examination alone, and confirming a diagnosis of malignant PCC necessitates the identification of distant metastasis [26]. Consequently, adrenal neoplasm presents the intricate challenge of prognostic prediction for these tumors.

Epithelial membrane proteins (EMPs) 1, 2, and 3 are members of the myelin protein 22-kDa (PMP22) gene family. While they primarily function in the peripheral nervous system, their distinct roles in various tumors have been documented [27,28]. EMP 1 is involved in tumor cell adhesion through the PI3K/AKT pathway [29], EMP 2 contributes to tumor cell migration via the FAK/Src pathway [30,31], and EMP 3 plays a role in tumor cell survival and metastasis through the ErbB2-PI3K-AKT pathway [32,33]. However, these EMPs have been reported to exhibit both tumor-promoting and tumor-suppressing effects in different types of cancers. For instance, EMP 1 is associated with metastatic features in melanoma [34], but acts as a negative regulator of tumor cell growth and metastasis in nasopharyngeal [35], stomach [36], and colorectal cancer [37]. EMP 2 functions as an oncogene in hormone-related cancers such as breast, ovarian, and endometrial cancer [31,38,39,40], while serving as a tumor-suppressor gene in nasopharyngeal [41] and urothelial cancer [42]. EMP 3 is upregulated in HER-2-positive breast cancer [43] and is associated with Myc proteins [44], and its knockdown has been shown to reduce cell proliferation and invasiveness in hepatocellular carcinoma [33]. Conversely, EMP 3 is downregulated in neuroblastoma [45], glioma [45], non-small cell lung cancer (NSCLC) [46], and esophageal cancer [47], where it inhibits cell proliferation. Although EMPs have been extensively studied in various tumor types, research on their expression in adrenal neoplasms has been limited. In the case of ACN, due to the limitations of predicting tumor behavior solely based on pathological findings, various diagnostic systems such as the Weiss scoring system [48], Reticulin algorithm [49], and Helsinki score [50] have been proposed and utilized to overcome this. Additionally, ancillary immunohistochemical markers such as Ki-67 [51], p53 [52], CYP11B-2 [53], SF-1 [54], CYP2W1 [55], and RRM1 [56] have been suggested as prognostic and/or predictive factors. However, their practical application in clinical settings is limited. Similarly, for PCC, relying solely on pathological observations for predicting tumor behavior has limitations. To address this, various diagnostic systems such as the PASS system [57], GAPP system [58], and COPPS system [59] have been proposed and used. Ancillary immunohistochemical markers such as Ki-67 [60], SDHB [61], and S-100 [62] have been suggested as prognostic factors. If the expression of EMPs is observed in specific adrenal tumors and the pattern of expression varies according to tumor biology, there is potential for EMPs to serve as prognostic and/or predictive factors in adrenal tumors. Thus, the objective of this study is to investigate the expression of EMP 1, 2, and 3 in human adrenal gland tumors and explore their potential implications.

## 2. Results

### 2.1. Patient’s Basal Characteristics

The basal characteristics of the patients included are presented in Appendix A. The study included a total of 132 cases of ACN, with 115 cases classified as ACA, 17 cases as ACC, and 189 cases as PCC. A comparison between ACC and ACA revealed that ACC exhibited a larger tumor size, higher Fuhrman grade, increased and atypical mitosis, and lower clear-cell proportion, demonstrating statistical significance (*p* < 0.001). Additionally, ACC displayed features such as diffuse architecture, necrosis, and invasion of venous/sinusoidal structures and capsules. Although two ACC cases had Weiss scores of 4 or less, they were diagnosed as ACC due to the presence of distant metastasis at the time of diagnosis. Notably, tumor recurrence, distant metastasis, and patient mortality were exclusively observed in ACC cases. Regarding PCC, the GAPP score ranged from 0 to 2 in 138 cases (73.0%), 3 to 6 in 50 cases (26.5%), and 7 to 10 in 1 case (0.5%). Tumor recurrence occurred in 5 cases (2.6%), distant metastasis in 7 cases (3.7%), and patient death in 11 cases (5.8%).

### 2.2. Expression of EMP 1, 2, and 3 in Adrenal Cortical Neoplasm and Pheochromocytoma

The immunohistochemistry results for EMP 1, 2, and 3, including H-scores, are shown in Table 1. The H-scores (mean ± SD, range) of EMP 1, 2, and 3 in ACN were 91.6 ± 101.0 (0–300), 2.2 ± 12.0 (0–90), and 63.4 ± 82.4 (0–300), respectively, while those in PCC were 105.9 ± 78.4 (0–300), 0.6 ± 5.6 (0–60), and 102.5 ± 80.1 (0–300), respectively (Table 1). Therefore, in ACN, EMP 1, 2, and 3 were defined as low when the H-score was ≤90, ≤2 and ≤65, respectively. In PCC, EMP 1, 2, and 3 were defined as low when the H-score was ≤100, ≤0, and ≤100, respectively. When investigating the expression of the EMP family in ACN and PCC, an examination of the IHC proportion score based on the IHC intensity score reveals a significant increase in the IHC proportion score as the IHC intensity score increases for EMP1 and EMP3 (Appendix A).

When examining the H-scores of EMP 1, 2, and 3 in relation to ACN and PCC, a statistically significant distinction was observed in EMP 3 (*p* < 0.001). Specifically, PCC exhibited significantly higher H-scores compared to ACN (Table 1 and Figure 1). After comparing the expression of EMP 1, 2, and 3 between ACA and ACC, a statistically significant difference was observed in the H-scores of ACC and ACA in EMP 1 (*p* < 0.001) and EMP 3 (*p* < 0.001), with ACC showing higher H-scores than ACA (Appendix A). Upon investigating the expression of EMP 1, 2, and 3 in low- and high-expression groups within ACA and ACC, a statistically significant distinction was observed in EMP 1 (*p* < 0.001) and EMP 3 (*p* < 0.001). Notably, ACC exhibited a higher proportion of high expression compared to ACA (Table 2 and Figure 2). In ACN, serum aldosterone levels showed differences according to EMP 1 and EMP 2 statuses. In the EMP 1 and EMP 2 low-expression groups, serum aldosterone levels were significantly elevated (*p* = 0.001 and *p* = 0.008, respectively, Appendix A).

When examining the H-scores of EMP 1, 2, and 3 in PCCs based on the GAPP score, a statistically significant distinction was observed in EMP 1, indicating higher H-scores in PCC cases with a GAPP score of 3 or higher (*p* = 0.018, Appendix A). Upon investigating the expression of EMP 1, 2, and 3 in low- and high-expression groups based on the GAPP score, a statistically significant distinction was observed in EMP 1. Notably, a higher proportion of PCCs with a GAPP score of 3 or more exhibited high expressions of EMP 1 (*p* = 0.013, Table 3). In PCC, there was no significant difference observed in 24 h urine catecholamine levels according to EMP-level status (Appendix A).

### 2.3. Correlations between EMP 1, 2, and 3 Expressions and the Clinicopathological Factors of Adrenal Neoplasm

Figure 3 displays the expression levels of EMP 1, 2, and 3 and their association with the clinicopathological factors of PCC and ACN. In PCC, the catecholamine type was associated with EMP 1 (*p* = 0.002), and the norepinephrine type was associated with high expression levels of EMP 1 (Figure 3). In ACN, EMP 1 and EMP 3 were associated with changes in mitosis (*p* < 0.001), atypical mitosis (*p* < 0.001), clear-cell proportion (*p* < 0.001), diffuse architecture proportion (*p* < 0.001), necrosis (*p* < 0.001), and the Weiss score (*p* < 0.001). High expression levels of EMP 1 and 3 were associated with higher levels of mitosis, atypical mitosis, lower clear-cell proportions, higher diffuse architecture proportions, an increase in necrosis, and higher Weiss scores when compared to that of low levels of EMP 1 and 3 expressions (Figure 3).

### 2.4. The Impact of EMP 1, 2, and 3 Expression Levels in PCC and ACN on Patient Prognosis

No statistically significant differences were found when analyzing the impact of EMP 1, 2, and 3 expression levels on patient prognosis in PCC using univariate analysis (Table 4). However, in ACN, the association between EMP 1, 2, and 3 expression and patient prognosis was investigated, and high levels of EMP 1 and 3 expressions were significantly associated with shorter overall survival (OS) (*p* = 0.001, Table 5 and Figure 4) when compared to low levels of EMP 1 and 3. In the multivariate Cox analysis, venous invasion (hazard ratio: 193.9, 95% CI: 3.054–12,313, *p* = 0.013) was the only factor that showed a significant association with shorter OS (Table 6).

## 3. Discussion

This study focused on examining the expression of EMPs in tumors originating from the adrenal gland. Initially, it was observed that the expression of EMP 3 was elevated in PCC compared to ACN. Previous studies have examined the expression of EMP 1, 2, and 3 in various human tumors, but there are limitations to direct comparisons with previous studies as there has been no investigation of EMP expression in ACN and PCC. It has been reported that EMPs may exhibit both tumor-suppressor and -promoter roles, and that their expression can increase or decrease depending on the type of tumor. For EMP 1, representative cancers demonstrating an increase in expression include head/neck [63,64], breast [65,66], and stomach cancer [67,68], while representative cancers exhibiting a decrease in expression include oral cavity [63], and nasopharynx cancer [35]. For EMP 2, representative cancers displaying an increase in expression include nasopharynx cancer [69,70] and uterine endometrial carcinoma [71,72], while those showing a decrease in expression include urothelial carcinoma [42]. Finally, for EMP 3, an increase in its expression is demonstrated in breast cancer [73], and a decrease in its expression is observed in lung cancer [46].

In PCC, the expression of EMP 3 was higher compared to that of ACN. A previous study suggested that EMP 3 (a myelin-related gene located at 19q13.3) is a likely tumor-suppressor gene, because of its genomic deletion in the 19q13 chromosomal region in neural origin tumors, such as neuroblastoma and glial tumors [74,75]. Further investigation has shown that EMP 3 transcriptional silencing occurs in neuroblastoma and glial tumors due to hypermethylation [45]. Therefore, as PCC is also a neural crest origin tumor like neuroblastoma, hypermethylation-mediated EMP 3 silencing can be expected in PCC. However, it has been reported that EMP 3 is more frequently methylated in neuroblastoma than in PCC (methylation rate: 68.4% versus 6.1%), which could explain the retainment of EMP 3 expression in PCC [76].

In PCC, there was a significant increase in the expression of EMP 1 when the GAPP score was three or higher. EMP 1 expression is commonly detected in early and immature neurons, indicating its potential association with neurogenesis during the development of both the central and peripheral nervous systems [77]. Since a high GAPP score in PCC indicates a higher malignant and metastatic potential [60,61], high EMP 1 expression in PCC could be associated with this potential. EMP 1 has been suggested to play a tumor-promoter role in other tumors by promoting cell proliferation, migration, and invasion [29,78,79,80,81,82]. The PI3K/AKT pathway is crucial in EMP 1’s oncogenic role [29,82]. In PCC, AKT signaling has been reported to be activated in various situations in in vitro cell line studies [83,84,85], and the expression of PI3K/AKT-pathway-related molecules is also reported to be high in human PCC tissues [83,86]. Moreover, the expression of phosphorylated S6, one of the PI3K/AKT-pathway-related molecules, is higher in metastatic PCC than in the non-tumor adrenal medulla and primary pheochromocytoma [87]. Therefore, further studies are necessary to investigate the impact of EMP 1 on the PI3K/AKT pathway and the malignant/metastatic potential in PCC. Various scoring systems have been proposed to predict the tumor behavior of PCC. Among these, the GAPP and COPPS scoring systems include ancillary IHC markers as components. In the GAPP system, Ki-67 is incorporated [58], while the COPPS system includes S-100 and/or SDHB [59]. Therefore, further research is needed to develop an effective system for predicting PCC behavior using a combination of multiple clinicopathologic factors and EMPs, particularly EMP1.

Our results suggest that EMP 1 and 3 may contribute to the oncogenic role in malignant tumors in ACN, as they showed significantly higher expression in ACC than in ACA. As a possible mechanism of action for EMP 1 and 3 in ACC, cross-talk with the ErbB family receptors (ErbB-1 (HER1 or epidermal growth factor receptor, EGFR), ErbB-2 (HER2), ErbB-3 (HER3), and ErbB-4 (HER4)) may exist. Previous studies have reported functional interactions between EM 3 and HER-2 in urothelial and breast cancer [32,43,88], and further interplay between EMP 1 and EGFR in lung cancer [89]. ACC also showed significant EGFR overexpression when compared to ACA [90,91], indicating that increased EMP 1 and EMP 3 may interact with EGFR. Additionally, one of the signaling pathways activated in ACC is IGF-IGFR signaling [92], with co-overexpression of EGFR and IGF1R observed in approximately 53% of ACC cases [91]. Therefore, further studies are needed to investigate the interaction between EMP 1/EMP 3 and cell membranous receptors such as EGFR and IGF1R in ACC. Another possible mechanism for the oncogenic role of EMP 1 and EMP 3 in ACC, is the activation of the PI3K/AKT pathway. ACC showed higher expression of *p*-Akt (Ser473), a molecule related to the PI3K/AKT pathway, than that of ACA and normal tissue [93,94]. Since EMP 1 and 3 play an important role in the PI3K/AKT pathway in other tumors [29,32,33], their association with the PI3K/AKT pathway in ACC can also be proposed. However, additional studies are needed to investigate this. In this study, high expression of EMP 1 and 3 was associated with poor prognosis in ACN. Previous studies have also reported that high expression levels of EMP 1 are a poor prognostic factor in urothelial carcinoma [95] and pediatric leukemia [79]. Furthermore, high expression levels of EMP 3 are a poor prognostic factor in brain glioma [62], breast phyllodes tumor [96], gastric cancer [97], and urothelial carcinoma [32]. However, EMPs have dual roles as tumor-suppressors and tumor-promoters. Therefore, additional studies are needed to determine their role as tumor prognostic markers depending on the type of tumor. Various scoring systems have been proposed to predict the tumor behavior of ACC. Among these, the Reticulin algorithm and Helsinki score system include ancillary special stain results as components. In the Reticulin algorithm, reticulin stain is incorporated [49], while the Helsinki score system includes Ki-67 [50]. Therefore, further research is necessary to investigate an effective system for predicting ACC behavior using a combination of multiple clinicopathologic factors and EMPs, especially EMP1 and EMP3.

This study suggests that EMP may be a therapeutic target for adrenal neoplasms such as ACC and PCC. Previous studies have demonstrated the inhibitory effects of anti-EMP 2 recombinant bivalent antibody fragments (diabodies) on the proliferation and induction of apoptosis in uterine endometrial and ovarian cancer [39,98]. Moreover, anti-EMP 2 IgG1 was shown to promote cell death and inhibit cell invasion in breast cancer [40]. Therefore, EMP inhibitors could be proposed as one of the therapeutic agents for tumors, but there are several obstacles to developing monoclonal antibodies that target EMP. One of the most important obstacles is that EMP has a very complex effect on tumors, showing different tumor-suppressor and -promoter roles depending on the distinct tumor type. Therefore, further preclinical and clinical studies targeting adrenal neoplasms are needed.

In conclusion, EMP expression levels were shown to be significantly different between ACN and PCC, and EMPs are associated with malignant tumor biology in adrenal cortical neoplasm and pheochromocytoma, suggesting the role of a prognostic and/or predictive factor for EMPs in adrenal tumor.

## 4. Materials and Methods

### 4.1. Patient Selection

The focus of this research was on individuals who underwent surgery for ACC and PCC between January 2000 and December 2012 at Severance Hospital. The study did not include individuals who underwent preoperative chemotherapy. The Institutional Review Board of Yonsei University Severance Hospital granted approval for the study (IRB number: 4-2021-0393). Endocrine pathologists (Koo JS) retrospectively reviewed all cases, conducting histology using hematoxylin and eosin (H&E)-stained slides. Clinicopathological data, such as age at diagnosis, disease recurrence, metastasis, current status, and length of follow-up, were extracted from the patients’ medical records.

### 4.2. Tissue Microarrays

After careful selection of representative regions on H&E-stained slides, corresponding spots were identified on the surface of the corresponding paraffin block. Core biopsies measuring 5 mm were then obtained from the chosen regions and placed into a recipient block measuring 5 × 4. To reduce extraction bias, more than two tissue cores were taken from each case. Each tissue core was assigned a distinct microarray location number, which was linked to a comprehensive database containing additional clinicopathological information.

### 4.3. Immunohistochemistry

In this research, immunohistochemistry (IHC) was carried out utilizing formalin-fixed, paraffin-embedded tissue sections. Tissue sections measuring 3 μm in thickness were prepared from paraffin blocks. These sections underwent a process of deparaffinization and rehydration using xylene and alcohol solutions. For antigen retrieval, cell conditioning 1 (CC1) buffer (citrate buffer pH 6.0, Ventana Medical System) was employed. IHC staining was conducted using the Ventana Discovery XT automated stainer (Ventana Medical System, Tucson, AZ, USA), which included appropriate positive and negative controls. The antibodies employed for IHC in this study are listed in Appendix A and IHC stains for EMP 1, EMP2 and EMP3 in normal adrenal gland are shown in Appendix A.

### 4.4. Interpretation of Immunohistochemical Staining

Immunohistochemical markers were visualized using light microscopy. The expression of these markers was assessed using the semi-quantitative H-score method and evaluated in tumor cells. The H-score system generates a comprehensive range from 0 to 300, which is determined by multiplying the dominant staining intensity score (0 = no staining; 1 = weak or barely detectable staining; 2 = distinct brown staining; 3 = strong dark brown staining) by the percentage (0–100%) of positive cells [99]. If the calculated H-score exceeded the mean value, it was categorized as high expression; conversely, if it fell below the mean value, it was classified as low expression.

### 4.5. Statistical Analysis

The Statistical Package for the Social Sciences (SPSS) for Windows, Version 24.0 (SPSS Inc., Chicago, IL, USA) was utilized for data analysis. To determine statistical significance, Student’s *t*-test and Fisher’s exact test were employed for continuous and categorical variables, respectively. In the case of analyzing data with multiple comparisons, a corrected *p*-value was calculated using the Bonferroni multiple comparison procedure. Statistical differences were considered significant if *p* < 0.05. Kaplan–Meier survival curves and log-rank statistics were employed to evaluate tumor recurrence time and overall survival (OS). Additionally, the Cox proportional hazards model was utilized to perform multivariate regression analysis.

## Figures and Tables

**Figure 1 ijms-24-13016-f001:**
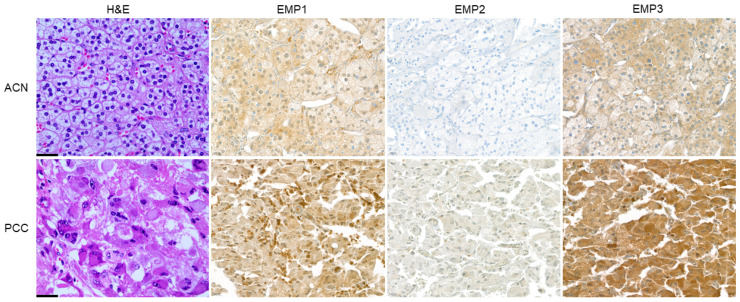
Expression of epithelial membrane proteins (EMPs) 1, 2, and 3 in adrenal neoplasm. Pheochromocytomas (PCC) show higher expression of EMP 3 than that of adrenal cortical neoplasm (ACN) [X400, scale bar: 20 μm].

**Figure 2 ijms-24-13016-f002:**
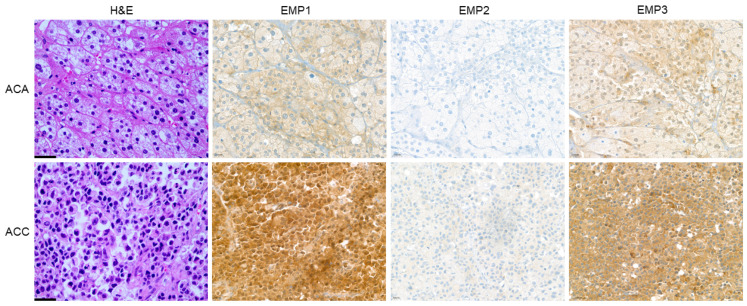
Expression of EMP 1, 2, and 3 in adrenal cortical neoplasm. Adrenal cortical carcinomas (ACC) show higher expression of EMP 1 and 3 than that of adrenal cortical adenomas (ACA) [X400, scale bar: 20 μm].

**Figure 3 ijms-24-13016-f003:**
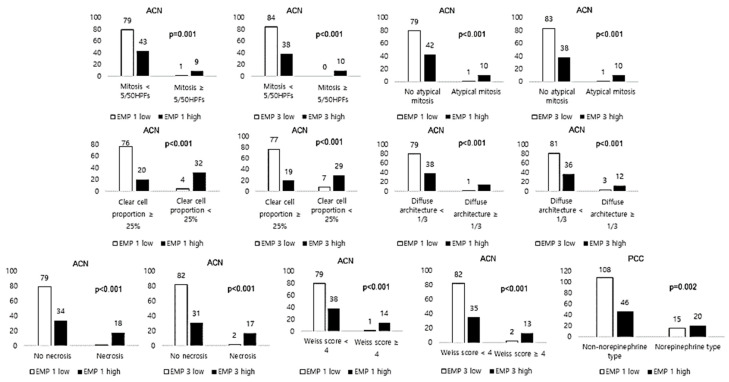
Correlations between EMP 1, 2, and 3 expressions and the clinicopathological factors of adrenal neoplasm.

**Figure 4 ijms-24-13016-f004:**
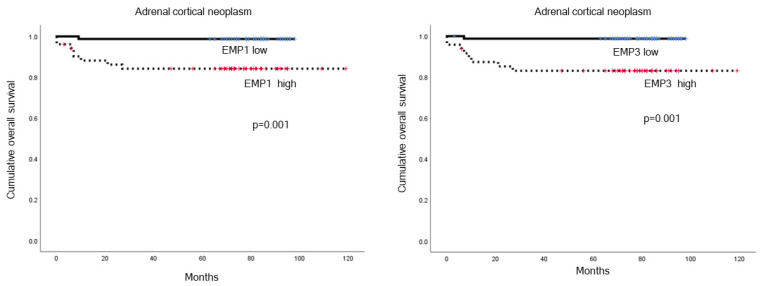
The impact of EMP 1 and 3 expressions in ACN on patient prognosis. EMP 1 and 3 were found to be significantly associated with overall survival (OS), with high EMP 1 and 3 expression levels being associated with shorter OS in adrenal cortical neoplasm (*p* = 0.001).

**Table 1 ijms-24-13016-t001:** H-scores of EMP 1, 2, and 3 in adrenal neoplasm.

EMP Type	Total *N* = 321 H-Score (Mean ± SD)	Adrenal Cortical Neoplasm *n* = 132 H-Score (Mean ± SD)	Pheochromocytoma *n* = 189 H-Score (Mean ± SD)	*p*-Value
EMP1	100.0 ± 88.5	91.6 ± 101.0	105.9 ± 78.4	0.154
EMP2	1.3 ± 8.8	2.2 ± 12.0	0.6 ± 5.6	0.134
EMP3	86.4 ± 83.2	63.4 ± 82.4	102.5 ± 80.1	<0.001

SD, standard deviation.

**Table 2 ijms-24-13016-t002:** Expression of EMP 1, 2, and 3 in adrenal cortical neoplasm.

EMP Type	Total *N* = 132 (%)	Adrenal Cortical Adenoma, *n* = 115 (%)	Adrenal Cortical Carcinoma, *n* = 17 (%)	*p*-Value
EMP 1				<0.001
Low	80 (60.6)	79 (68.7)	1 (5.9)	
High	52 (39.4)	36 (31.3)	16 (94.1)	
EMP 2				0.594
Low	125 (94.7)	108 (93.9)	17 (100.0)	
High	7 (5.3)	7 (6.1)	0 (0.0)	
EMP 3				<0.001
Low	84 (63.6)	82 (71.3)	2 (11.8)	
High	48 (36.4)	33 (28.7)	15 (88.2)	

**Table 3 ijms-24-13016-t003:** The expression levels of EMP 1, 2, and 3 in pheochromocytoma based on the GAPP score.

EMP Type	Total *N* = 189 (%)	GAPP < 3 *n* = 138 (%)	GAPP ≥ 3 *n* = 51 (%)	*p*-Value
EMP 1				0.013
Low	123 (65.1)	97 (70.3)	26 (51.0)	
High	66 (34.9)	41 (29.7)	25 (49.0)	
EMP 2				0.565
Low	186 (98.4)	135 (97.8)	51 (100.0)	
High	3 (1.6)	3 (2.2)	0 (0.0)	
EMP 3				0.728
Low	126 (66.7)	91 (65.9)	35 (68.6)	
High	63 (33.3)	47 (34.1)	16 (31.4)	

GAPP, grading system for adrenal pheochromocytoma and paraganglioma.

**Table 4 ijms-24-13016-t004:** The impact of EMP 1, 2, and 3 expressions in pheochromocytoma on disease-free survival and overall survival, assessed through univariate analysis using the log-rank test.

EMP Type	No. of Patients/ Recurrence/Death	Disease-Free Survival		Overall Survival	
		Mean Survival Months (95% CI)	*p*-Value	Mean Survival Months (95% CI)	*p*-Value
EMP 1			0.468		0.564
Low	122/4/6	151 (140–162)		153 (142–164)	
High	66/1/5	154 (148–161)		141 (126–155)	
EMP 2			0.727		0.645
Low	185/5/11	N/A		N/A	
High	3/0/0	N/A		N/A	
EMP 3			0.550		0.784
Low	125/4/8	151 (142–161)		150 (138–161)	
High	63/1/3	102 (99–105)		97 (91–104)	

CI, confidence interval.

**Table 5 ijms-24-13016-t005:** The impact of EMP 1, 2, and 3 expressions in adrenal cortical neoplasm on disease-free survival and overall survival, assessed through univariate analysis using the log-rank test.

EMP Type	No. of Patients/ Recurrence/Death	Disease-Free Survival		Overall Survival	
		Mean Survival Months (95% CI)	*p*-Value	Mean Survival Months (95% CI)	*p*-Value
EMP 1			0.019		0.001
Low	80/0/1	N/A		96 (94–99)	
High	52/3/8	N/A		101 (90–112)	
EMP 2			0.669		0.467
Low	125/3/9	N/A		N/A	
High	7/0/0	N/A		N/A	
EMP 3			0.013		0.001
Low	84/0/1	N/A		96 (94–99)	
High	48/3/8	N/A		100 (88–112)	

CI, confidence interval.

**Table 6 ijms-24-13016-t006:** Multivariate overall-survival analysis of patients with adrenal cortical neoplasm.

Parameter	Hazard Ratio	95% CI	*p*-Value
Fuhrman grade			0.507
1, 2 versus 3, 4	5.070	0.042–613.7	
Mitosis (/50HPFs)			0.401
≤5 versus >5	4.320	0.142–131.8	
Atypical mitosis			0.318
Absent versus Present	3.671	0.287–47.04	
Clear cell proportion			0.293
≥25% versus <25%	9.681	0.140–668.3	
Diffuse architecture (proportion)			0.430
<1/3 versus ≥1/3	3.092	0.188–50.97	
Venous invasion			*0.013*
Absent versus Present	193.9	3.054–12,313	
Capsular invasion			0.950
Absent versus Present	1.075	0.110–10.48	
Weiss score			0.384
<4 versus ≥4	0.101	0.001–17.62	
EMP 1			0.244
Low versus High	20.892	0.126–3458	
EMP 3			0.475
Low versus High	8.658	0.023–3210	

CI, confidence interval.

## Data Availability

All data pertaining to the study are comprehensively included in both the article and its Appendix A.

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
