# Peer review of "Expression of EMP 1, 2, and 3 in Adrenal Cortical Neoplasm and Pheochromocytoma"

_ijms, 2023, doi:10.3390/ijms241613016_

Round 1
Reviewer 1 Report
The Authors present the results of a study investigating the expression of EMP in adrenal gland neoplasm and pheochromocytoma.
The study was well conducted and the quality of presentation is high. However, the reader can't immediately get the potential relevance of the findings. This is particularly evident in the abstract, which leaves a feeling of 'so what'. I feel that also the Introduction and the Discussion do not fully clarify the rationale of the study and the relevance of the findings.
Can the Authors work on this?
No special concerns.
Author Response
[ANSWER] In this study, the significance of EMP results primarily lies in its role as a prognostic and/or predictive factor in adrenal tumors. We elaborated on this aspect in the abstract, introduction, and discussion sections. (page 1 line 20-22, page 2 line 72-85, page 8 line 232-237, page 8 line 263-268).
Reviewer 2 Report
Manuscript entitled "Expression of EMP 1, 2, and 3 in adrenal cortical neoplasm and pheochromocytoma". I appreciate the large cohort and solid study.
Major issues:
1. The higher magnification of IHC should be provided to confirm the staining pattern is correct.
2. The staining pattern of normal tissue should also be provided.
3. The expression of EMPs should be correlated with hormone levels.
4. Cases with bland morphology and with frank malignant histologic type should be shown.
Acceptable
Author Response
- The higher magnification of IHC should be provided to confirm the staining pattern is correct.
[ANSWER] I have updated the image to a high-resolution photo that is magnified by a factor of 400.
- The staining pattern of normal tissue should also be provided.
[ANSWER] I have included a photograph of normal tissue in Supplementary Figure 1.
- The expression of EMPs should be correlated with hormone levels.
[ANSWER] In ACN patients, serum aldosterone levels were added for EMPs expression and analysis, while in PCC patients, 24-hour urine catecholamine levels were included (page 3 line 127-130, page 4 line 146-147).
- Cases with bland morphology and with frank malignant histologic type should be shown.
[ANSWER] H&E images of clearly distinguishable cases of benign and malignant have been added.
Round 2
Reviewer 2 Report
The revision is acceptable.
Acceptable